# The Health and Nutritional Status of Children (10–18 years) Belonging to Food Insecure Households: The Korea National Health and Nutrition Examination Survey (2012–2019)

**DOI:** 10.3390/ijerph20176695

**Published:** 2023-08-31

**Authors:** Sowon Jung, Jieun Shin, Myoungsook Lee

**Affiliations:** 1Department of Food & Nutrition, Sungshin Women’s University, Seoul 01133, Republic of Korea; sw7142@hanmail.net; 2Healthcare Data Science Center, Bio-Medical Informatics, College of Medicine, Konyang University, Daejeon 35365, Republic of Korea; jeshin@konyang.ac.kr; 3Research Institute of Obesity Science, Sungshin Women’s University, Seoul 01133, Republic of Korea

**Keywords:** food-secure household, food insecure household, children and adolescents, clinics, lifestyles, nutrition

## Abstract

To research the health and nutritional status in Korean children and adolescents belonging to food insecure households (FI), the preregistered secondary data of 18 items from the Food Security Evaluation in the Korea National Health and Nutrition Examination Survey (KNHANES; 2012–2019) were used. Comparative analyses (food security group [FS], (*n* = 3150) vs. FI, (*n* = 405) of household characteristics, health status (anthropometrics, clinics, mentality), and nutritional status (nutrient intake, diet-quality, and pattern) were performed in children (boys: 1871, girls: 1684) aged 10–18 years. The FI comprised higher proportions of participants from low-income families, basic livelihood-security recipients, and vulnerability (characteristics: female household heads, aged ≥50, single, unemployed, with low education and unmet healthcare needs). Compared to FS, boys had higher abdominal obesity and alcohol use, whereas girls had lower high-density-lipoprotein cholesterol (HDLc) and mental vulnerability (self-perceived obesity despite FS-similar anthropometry) in FI. Inadequate protein intake among boys and girls, and high carbohydrate and inadequate fat intake among girls were especially found in the FI status. From the results of a nutrition quality test, Vit-A in boys, and protein, niacin, and iron intakes in girls were insufficient, respectively. Health-nutritional policies to improve children’s lifestyles should reinforce FI-based intake of deficient nutrients.

## 1. Introduction

Food security (FS) is defined as a scenario wherein individuals have access to nutritionally adequate, safe food through socially acceptable means to facilitate energetic, healthy living. In contrast, food insecurity (FI) is a situation characterized by a lack of access to secure nutritionally adequate and safe food through socially acceptable means [1]. FS assessment reflects food availability, accessibility, utilization, and stability, wherein a lack of any one of these is considered to indicate the absence of FS [2]. The Korean Household Food Security Survey Module (K-HFSS), based on the US Household Food Security Survey Module (US-HFSS), has been used since the 5th KNHANES in 2012 [3,4]. In South Korea, which is currently ranked 10th globally, owing to its economic advances, income-based social polarization persists despite the better health status of the general population [5,6]. Data from 2019 Statistics, FI affects 13.0% of households in the “low” income bracket, and it was 3.7 times higher than the national average of 3.5% in Korea [7].

FI is adversely associated with physical and mental health of adults and children without racial differences, however, extensive research on negative effects of FI for children was necessary. FI experiences in children eventually lead to disparities in adulthood, presenting health risk factors, and therefore emphasize the need for government systems. Diverse factors associated with FI-households were observed in children and adolescents (children) compared to adults, such as stunted growth, lower education level, poor health status, and a low quality of nutrition, because of unstable socioeconomic status of FI housing [8,9]. In the USA, individuals who experience severe FI face a 20% higher 10-year risk of cardiovascular diseases compared to their FS counterparts [10,11]. From US-HFSS, 14.7% of US households were at the risk of FI, and FI experiences in childhood and adolescents are associated with increased risk of physiological, psycho-emotional, social, and academic developments in the USA [12,13,14,15]. The Canadian National Longitudinal Survey of Children and Youth linked FI to stress, anxiety, social agitation, and chaotic household dynamics, and in the US National Longitudinal Study of Adolescent to Adult Health, children and adolescents with FI were particularly vulnerable to psychological problems, such as anxiety and depression [16,17]. In the systematic review and meta-analysis, FI increased the risk of stunting and being underweight in children living in developing countries, compared to developed countries [18]. Children from FI households skip breakfast often, have lower milk intake, and have higher consumption of sugar-sweetened beverages and they are more likely to experience dietary diversity or nutritional deficiencies [9,19]. In a US high school survey, adolescents with FS were more than twice as likely to not eat breakfast (OR = 2.27, 95% confidential interval [CI]: 1.61–3.21), and current smoking (OR = 1.65, 95%Cl 1.16–2.36). and current alcohol intake (OR = 1.36, 95%Cl 1.01–1.84) increased health risks [20]. Importantly, the coronavirus (COVID-19) pandemic led to households entering a severe FI status due to inadequate accessibility to healthy lifestyles, thus directly/indirectly affecting the nutritional health status of children and adolescents [21,22].

As the aim of this study is to disclose the nutritional health status of Korean children (10–18 years) who are the most vulnerable to FI during lifecycle, we undertook a comparative analysis of household characteristics/factors with FS or FI, such as anthropometrics, socioeconomics, clinics, mental health, dietary lifestyle, and nutritional quality. Our longitudinal report for Korean children with FI-households will facilitate the development of educational and policymaking programs, such as food support and nutritional management, and with these efforts, help to provide stability to the amount of vulnerable people within the FI population.

## 2. Materials and Methods

### 2.1. Research Design

The raw data from Year 3 (2012) of the 5th KNHANES, 6th KNHANES (2013–2015), and Year 1 (2019) of the 8th KNHANES were used. From a total of 41,127 potential participants (consisting of 10,069 from Year 3 of the 5th survey; 22,948 from the 6th survey; and 8110 from Year 1 of the 8th survey), 4048 children and adolescents aged 10–18 years (children), belonging to households—a population with a high rate of FI—were identified. After excluding 315 individuals who did not respond to the FS survey (18 items) and 178 with a daily energy intake less than 500 kcal or more than 5000 kcal, a total of 3555 children (1871 boys and 1684 girls) were enrolled in this study. The children were divided into the FS (*n* = 3150) and FI (*n* = 405) groups by Nikitto’s method, and the correlations among household characteristics, diet patterns, and health status were analyzed [23], as shown in Figure 1. All data were analyzed after complex sample weighting to ensure that the sample was representative of the population [23]. The review process of this study was exempted by IRB at Sungshin Women’s University (Review exemption #; SSWUIRB-2021-046) because we used the preregistered secondary data from KNHANES.

### 2.2. Tools for the Evaluation of FS and FI

The 18-item K-HFSS, which was modified and developed from the US-HFSS, was used to assess FS and FI. In the KNHANES, the FS survey deals with the standard of the diet at the household level, and two different methods (single question and 18 items of the K-HFSS) were used. In the present study, data from Year 3 of the 5th survey, the 6th survey, and Year 1 of the 8th survey were analyzed, and the 18-item FS survey were used. The 7th survey used only a single question about FS and was therefore excluded from our study. The total number of items depends on the inclusion of a child in the household, wherein households with a child have 18 items (including 8 children-specific items) and households without a child have 10 items. The FS score for households with a child are categorized as food secure (0–2), mildly food insecure (without hunger, 3–7), moderately food insecure (with hunger, 8–12), and severely food insecure (with hunger, 13–18) [24]. In the present study, the scores were divided into two categories according to the report by Nikitto et al.: food secure (FS: 0–2) and food insecure (FI: mildly, moderately, and severely food insecure, 3–18) [23] Table 1.

### 2.3. General Characteristics of the Participants

The sex- and age-stratified percentages of FS and FI groups were analyzed for the entire population and the child-included households. Differences in sex, area of residence, household income, national basic livelihood security (NBLS) beneficiary status, household size, and head of household characteristics (sex, age, marital status, education level, type of health insurance, private health insurance, health service utilization) according to FS status were analyzed. 

### 2.4. Health Status

Health-related characteristics included anthropometric measurements, blood test results, mental health, and health lifestyle. Anthropometric measurements included height, weight, body mass index (BMI), and waist circumference (WC); blood test parameters included glycated hemoglobin (HbA1c; %), fasting blood sugar (FBS), total cholesterol (TC), high-density lipoprotein (HDL), low-density lipoprotein (LDL), and triglyceride (TG). The obesity index was computed based on the age- and sex-specific BMI percentile standards defined in the 2017 Growth Chart for Children, which was provided by the Ministry of Health and Welfare and the Korean Pediatric Society, was as follows: <5% (underweight), 5–85% (normal weight), and >85% (overweight/obese).

Mental health and health lifestyle included subjective health status, subjective body image, smoking status, and drinking status. Subjective health status was classified as good, normal, and bad, and subjective body image was classified as lean, normal, and obese. Smoking status was classified as non-smoker, ex-smoker, and current smoker, and drinking status was classified as non-drinker (lifetime non-drinker or no alcohol in the past year) and drinker (alcohol consumption in the past year). 

### 2.5. Dietary Lifestyles and Food/Nutrition Intake

Dietary lifestyle was analyzed in terms of breakfast/lunch/dinner frequency in the past week, eating together with family or other people during breakfast/lunch/dinner, and dining-out frequency. Interest in nutritional facts was analyzed in terms of utilization of nutritional facts, nutrients of interest on nutritional facts, and being influenced by nutritional facts. 

Nutrient intake was compared using the 2010 Dietary Reference Intake for Koreans (KDRIs) for the 5th and 6th survey data and 2015 KDRIs for the 8th survey data [26,27]. As the 5th and 6th surveys used retinol equivalents (RE: retinol + 1/6 × beta-carotene) for vitamin A, whereas the 8th survey used retinol activity equivalents (RAE: retinol + 1/12 × beta-carotene) for vitamin A, the vitamin A values presented in the 5th and 6th surveys were converted to RAE for direct comparison across all data points [24]. Acceptable Macronutrient Distribution Ranges (AMDR) constitute the percentage of each energy source (carbohydrate, protein, and fat) from the total energy intake. The adequacy of intake was determined based on the AMDR presented in the 2015 KDRIs: 55–65% for carbohydrates, 7–20% for proteins, and 15–30% for fats. Nutrient intake was analyzed using the recommended nutrient intake (RI) as specified in the KDRI, based on the nutrient adequacy ratio (NAR), mean adequacy ratio (MAR), and index of nutrition quality (INQ) [28,29]. NAR was calculated “individual’s intake of a particular nutrient” divided by “recommended intake for the particular nutrient”. MAR was calculated the sum of NAR values for 9 nutrients (∑NAR) devided by 9. INQ was calculated “nutrient intake corresponding to 1000 kcal” devided by “recommended intake of the nutrient per 1000 kcal”.

To assess the quality of meals in the group, the percentage of individuals with insufficient nutrient intake (% Nutrition insufficient intake) was analyzed. Using the average requirement (EAR) cutoff method propounded by Beaton (1994), the ratio of nutrient intake (protein, vitamin A, thiamin, riboflavin, niacin, vitamin C, calcium, phosphorus, and iron) to the EAR was computed, and the percentage of individuals with intake below the EAR was estimated [30]. 

To evaluate the food groups, 22 food groups in the KNHANES were re-categorized into six food groups based on the 2015 KDRIs, such as group-I (Grain with 300 Kcal/serving), group-II (Meat, Fish, Eggs, and Legumes with 100 kcal/serving), group-III (Vegetables with 15 kcal/serving) group-IV (Fruits with 50 kcal/serving), group-V (Milk and Dairy Products with 125 kcal/serving) and group-VI (Fats and Sugars with 45 kcal/serving). The actual consumption frequency of each food group was calculated by dividing the energy intake of each food group by the serving size (kcal). The food group consumption frequency (%) was analyzed by comparing the actual consumption frequency of the six food groups with the Korean nutrient intake standards for children, specifically Meal Pattern A (consuming milk and dairy products twice a day) [31]. To examine the detailed consumption patterns of the six food groups, the energy intake (kcal) for each food group was analyzed. Out of the 22 food group items in the KNHANES, items with insufficient intake or low consumption frequency among children, such as condiments, miscellaneous (plants), miscellaneous (animal), and alcoholic beverages, were excluded, and the remaining 18 items were included in the analysis.

### 2.6. Statistical Analysis

For data processing and analysis, variables pertaining to the food group and head of household characteristics were classified using the Statistical Analysis System (SAS, version 9.4; SAS Institute Inc., Cary, NC, USA), and analyses were generally performed using the Statistical Package for Social Sciences 26.0 (SPSS Inc., Armonk, NY, USA). 

As the KNHANES uses a stratified cluster sampling, complex sample analyses, which include weights, stratification variables, and clustering variables, were used. Categorical variables (e.g., general characteristics, health-related characteristics, dietary lifestyle) were analyzed using chi-square test and presented as frequency and percentage. Continuous variables (e.g., food group and nutrient intake) were analyzed with the Student’s *t*-test and presented as mean and standard error (SE).

## 3. Results

### 3.1. General Characteristics in FS and FI

The proportion of FI-households with children was 12.53%, compared to children with FS-households (87.48%) and without gender differences. Regardless of the inclusion of a child in the household, there was a significantly higher percentage of females than males in the FI group. The percentage of the FI group in households with children is shown in Figure 2. There were 46.2% male and 53.8% female participants. By age group, the proportions of the participants were as follows: 15.4% in 1–9 years, 27.2% in 10–18 years, 8.8% in 19–29 years, 13.4% in 30–39 years, 20.3% in 40–49 years, 7.1% in 50–59 years, 3.7% in 60–69 years, and 4.1% in ≥70 years. In households with children, children (10–18 years) were identified as the most vulnerable population for FI. 

There were no significant differences between the FS and FI groups in the gender distribution and the percentage of urban and rural residence. However, as a result of showing that the food insecurity group was affected by socioeconomic factors, compared to the FS, the FI group had a higher percentage of individuals with a household income (quartile) classified as low or middle-low, past, and current NBLS beneficiaries, and belonging to a household of 1–2 or ≥6 individuals Table 2. Sex distribution and area of residence (urban, rural) were not associated with FI in households with children. Furthermore, in the FI group, the head of household characteristics were primarily female, ≥50 years of age, single (never married/divorced/separated/widowed), low-education level, unemployed, and with annual unmet healthcare needs. The most common reason for unmet healthcare needs was financial constraints.

### 3.2. Health Status in FS and FI

Compared to their counterparts in the FS group, boys in the FI group had lower WC and plasma TC and a significantly higher drinking rate, as per Table 3. Girls in the FI group had lower plasma HDL, and despite having no differences in their anthropometric measurements from those of girls in the FS group, they perceived themselves as obese. Furthermore, girls in the FI group had higher stress levels and had significantly higher rate of depressive mood for two or more consecutive weeks, and suicidal ideation in the past year, which indicated greater mental health vulnerability. Particularly, both boys and girls in the FI group had significantly higher rates of “moderate” or “poor” responses for subjective health status and significantly higher smoking rates compared to their counterparts in the FS group.

### 3.3. Dietary Lifestyles and Food/Nutrition Intake

Compared to their counterparts in the FS group, both boys and girls in the FI group more frequently skipped breakfast; moreover, girls skipped lunch more frequently, as shown in Table 4. Regarding the rate of eating meals for 5 days or more a week, both boys and girls in the FI group showed a lower rate of breakfast frequency compared to the FS group. However, there was no significant intergroup difference in eating with family or others, the frequency of dining-out, and interest in nutritional facts (utilization of nutritional facts and being influenced by nutrition facts).

Table 5 shows the results of the analysis of the quality of food and the nutrient intake. Although both boys and girls in the FI group were consuming adequate energy per AMDR, their protein intake was lower than that in the FS group Table 5. Particularly, girls in the FI group showed higher carbohydrate intake and significantly lower fat intake. In terms of the quality of nutrients, the NAR for niacin and iron were significantly lower among girls in the FI group than those in the FS group. The INQ was significantly lower for vitamin A among boys and for protein and niacin among girls in the FI group compared to their counterparts in the FS group. Furthermore, an INQ below 1 is considered to indicate insufficient nutrient intake, and both boys and girls in the FI group had insufficient intake of vitamin A, vitamin C, and calcium intake. Notably, girls in the FI group showed an insufficient intake of niacin and iron. Regarding the intake frequency (%) in the six food groups, boys in the FI showed significantly lower intake of Fats and Sugars as compared to their counterparts in the FS group, whereas girls showed significantly lower Fish and Meats group (meats, fish, eggs, legumes) intake compared to their counterparts in the FS Appendix A. There were no significant differences in the intake frequency of other food groups. Regarding the energy intake (kcal) from each food group, there was no significant difference among boys in the FI and FS; however, the girls in the FI showed significantly higher grain and seaweed intake and significantly lower potato/starch and fish and seafoods.

## 4. Discussion

This research was the first report to analyze the diverse factors, including health and nutritional status, in Korean children (10–18 years) who are the most vulnerable population for FI, using the 2012–2019 KNHANES data. During the 7 years (prior to COVID-19), children within FI-households had poorer environmental and lifestyle characteristics, such as skipping breakfast, drinking, and smoking, as well as insufficient consumption of proteins, niacin, iron, vitamin A and relevant food groups. 

Our results show that a high proportion of FI-households with children aged 10–18 years (27.2% of total) during the lifecycle are consistent with the findings of Kirkpatrick et al. and Hanson et al. [32,33]. FI-households with children showed characteristic features of vulnerable households, such as low income, past or current NBLS beneficiary status, household size of 1–2 or ≥6, low-education level of head of household, single (never married/divorced/separated/widowed) head of the household, unemployed head of household, and unmet healthcare needs of the head of the household. 

Diverse factors associated with FI household were observed in children, such as socioeconomic status, education level, unstable housing, employment challenges, health status, and nutritional quality [8,9]. According to the Canada Food Insecurity Report and the “Household Food Security in the United States 2020” published by the USDA Economic Research Service (ERS), the prevalence of FI was higher in low-income households with children [34,35]. In Korean studies, vulnerable populations, such as poor households, individuals with low education, individuals with disabilities, members of vulnerable households, divorced individuals, and women, are consistently facing a higher risk of FI [36,37,38]. Similar results were found in Nigerian survey, which evidenced that household food security status was highly vulnerable to income loss, and in the United States food insecurity study, FI households were associated with lower levels of education, lower incomes, and lower access to health care. In a Canadian–Quebec cohort study, the prevalence of household food insecurity was associated with larger households, single-parent households, lower income levels, and lower education levels with a greater risk of food insecurity [22,39,40]. Furthermore, heads of the household with FI showed a higher rate of unmet healthcare needs due to financial reasons. In the 2011–2012 National Health Interview Survey, US adults (18–64 years) in severe FI-households were shown to have to choose between managing their health conditions and reducing their food expenses due to limited financial support, which leads to unmet healthcare needs [41]. From US-HFSS, it was reported that 14.7% of US households were at the risk of FI (1 in 5 children), and FI experiences in childhood and adolescents are associated with an increased risk of physiological, psycho-emotional, social, and academic developments in the USA [12,13,14,15]. In the systematic review and meta-analysis, FI increased the risk of stunting (OR = 1.17; 95%CI:1.09–1.25) and becoming underweight (OR = 1.17; 95%CI:1.01–1.36) in children living in developing countries, such as Malaysia, Iran, Ethiopoa and Indonesia, compared to developed countries like the USA and Canada [18]. Prolongation of such situations may impede long-term health management and quality of life among children from FI-households. These results support that children’s health with FI-households are linked to parents’ socioeconomic, educational, and healthcare status.

For the clinic factors, boys in the FI-households had lower WC and plasma TC, whereas girls in the FI group had lower plasma HDL, without differences in anthropometric measurements compared to their counterparts in the FS. Tester et al. reported that the association between FI and obesity is unclear among children with which our findings match [42]. In contrast, Jefari et al. reported that FI may increase the risk for abdominal obesity in children, which our findings contradict [43]. Parker et al. found that FI households had a 1.65-fold increased chance of metabolic syndrome risk factors, such as high WC, BMI, FBS, and TG, but low HDLc, in low-income adults, but not in adolescents [44]. Fulay et al. found that there were no associations between and FI and cardiovascular risk, such as age-specific BMI-Z-scores, SBP/DBP, HDL-c, TC, FBS, TG, and LDL-c in family and children. Those results were consistently observed in our findings [45].

In terms of healthy lifestyle factors, there was a higher rate of smoking among boys and girls in the FI group compared to their counterparts in the FS group, and the rate of drinking was also significantly higher among boys. These results are in line with the findings of Sanjeevi et al., who reported that children from FI-households are at higher odds for tobacco exposure [46]. As study data indicate a higher correlation between FI and smoking exposure in children, continuous smoking management is crucial for this population in order to foster a healthy lifestyle. Regarding mental health, a high percentage of both boys and girls in the FI group perceived their health negatively as compared to their counterparts in the FS group. Several studies show that children and adolescents with food insecurity are vulnerable to psychological problems [16,17]. Additionally, students from food-insecure households were significantly more likely to have lower self-esteem and significantly lower global self-efficacy scores for making healthy choices than students from food-secured households [47]. In particular, girls demonstrated higher mental health vulnerability, as evidenced by perceiving themselves as obese despite no significant intergroup differences in their anthropometric measurements. In NHANES III, FI is strongly associated with depression and suicidal ideation in US adolescents (15–16 years), and girls are more likely than boys to exhibit a depressive mood, depressive disorder, and suicide [48]. Our findings confirm these results, but not their results pertaining to suicidal impulse.

In terms of dietary factors, both boys and girls in the FI group skipped breakfast more frequently than their counterparts in the FS group, as well as Nakitto’s study, which reported a higher rate of breakfast skipping in the FI group [24]. Skipping breakfast is a risk factor for eating disorders related to metabolic disorders and diseases such as depressive symptoms. Eating breakfast can have a positive effect on psychological well-being and reduce hunger in children and adolescents [49]. Although the AMDRs were met among both boys and girls in the FI group, their protein intake was lower than that in the FS group; of note, girls had higher carbohydrate but lower fat intake, as compared to the FS group. These findings are partially consistent with the results of Shim et al. and Park in which the FI group shows a higher carbohydrate intake and lower protein and fat intake, even with a similar calorie intake in adults [50,51].

In terms of the quality of nutrient intake (NAR and INQ), boys in the FI group had insufficient vitamin A intake, whereas girls in FI group had insufficient protein, niacin, and iron intake. According to Nakitto et al. and the “systematic review of the association between food insecurity and quality of meals in adults and children in the United States” by Hanson et al., FI-adults showed poorer meal quality; however, food-insecure children only had lower fruit intake and were not strongly associated poor meal quality [23,33], partially consistent with our findings pertaining to boys. Eicher–Miller et al. reported that the odds of iron-deficiency anemia in children aged 12–15 years from FI-households was almost 3.0 times higher compared to FS-households because of meal consumption with inadequate iron content [52]. Sachan et al. and Gonete et al. reported that there was a significant association between anemia and dietary diversity scores, FI, and living with at least one parent [53,54]; it was observed, both in developing and developed countries, with girls being at a higher risk for anemia due to an increased demand to compensate for blood lost through menstruation. Consistently, with our results pertaining to girls, iron deficiency warrants more attention, as it is linked to fatigue, learning disorders, and diminished productivity in adolescence, which is marked by growth spurts [9,55].

Since global economic growth and gross domestic products in 2020/2021 was 5–6.5% less than pre-coronavirus disease (COVID-19) in 2019, COVID-19 led households into severe FI status due to an inadequate accessibility to healthy lifestyles, thus directly/indirectly affecting the nutritional health status of children [21,22]. In a UK study, those experiencing the COVID-19 pandemic explained the less than 5% likelihood of being FI for single parents, but 30% not having access to healthy and nutritious food for young people aged 16–30 years [21]. Lee et al. reported that the rate of breakfast skipping increased in children since the COVID-19 pandemic, supporting that skipping breakfast is associated with the environmental factor of FI [56]. US policymakers reported that annual healthcare costs increased by $400 million dollars for every 1% increase in FI, therefore, rapid assessment methodology was applied to identify and address immediate needs among FI-children during COVID-19 [57].

This study has a few limitations. First, we used the data from the K-HFSS (18 items) for analysis. As a result, we had to exclude the seventh survey data that used a single item to assess food security, and this caused a gap in our examination of annual trends. Second, the K-HFSS contains sensitive questions about food expenses and is a self-reported survey. Therefore, the respondents may have been hesitant to respond accurately, which may have led to an underestimation of the prevalence of FI due to social acceptability bias. Third, examining a larger food-insecure population would produce adequate evidence pertaining to food intake in the food-insecure population, which in turn will enable a follow up of environmental factors (e.g., COVID-19) that affect FI, as well as the before-and-after studies.

## 5. Conclusions

Comparative analyses of household characteristics, health status, and nutritional status were performed in children belonging to a household with FS or FI status. The FI comprised higher proportions of participants from low-income families, basic livelihood-security recipients, and vulnerability. Compared to FS, boys had higher abdominal obesity and alcohol use, whereas girls had lower HDLc and mental vulnerability in FI. Inadequate protein intake in boys and girls, and high carbohydrate and inadequate fat intake in girls were especially found in the FI status. From the results of quality in nutrition, insufficient Vit-A among boys, and protein, niacin, and iron intake among girls were revealed. The findings of this study regarding the health and nutrition of Korean children with FI-households will facilitate the development of education and policymaking programs to promote health and nutritional well-being.

## Figures and Tables

**Figure 1 ijerph-20-06695-f001:**
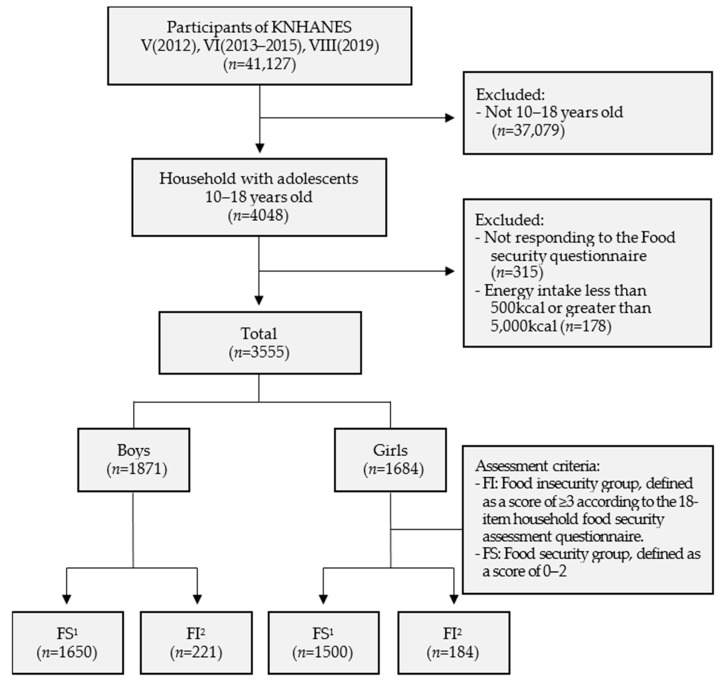
Flowchart for experimental design and subjects involved in this study. FS^1^: Food security group, FI^2^: Food insecurity group.

**Figure 2 ijerph-20-06695-f002:**
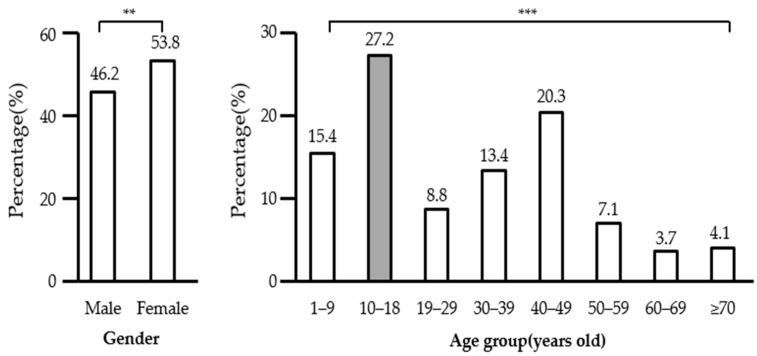
The prevalence (%) of food insecurity households with children (2012–2019, *n* = 32,659) according to gender and age groups with statistical significance, ** *p* < 0.01, and *** *p* < 0.001.

**Table 1 ijerph-20-06695-t001:** The 18 items of K-HFSS questionnaires in KNHANES and the evaluation of FS and FI [24,25,26].

Food Security Measures for KNHANES (18 Items)	Assessment		
Items for adult in household		Households with Children (Score)	Group
1. Food bought did not last and we did not have money to get more			
2. Worried food would run out before we got money to buy more			
3. Could not afford to eat balanced meals	Food secure	0–2	FS
4. Adult skipped meals (experience)	Food insecure (mildly) ^1^	3–7	FI
5. Adult skipped meals (frequency)			
6. Adult Cut size of meals	Food insecure (moderately) ^2^	8–12	
7. Respondent hungry but did not eat because could not afford			
8. Respondent lost weight	Food insecure (severely) ^2^	13–18	
9. Adult did not eat for whole day (experience)			
10. Adult did not eat for whole day (frequency)	Total	18	
Items for children in household			
11. Relied on few kinds of low-cost food to feed children			
12. Could not feed children balanced meals			
13. Children were not eating enough			
14. Cut size of children’s meals			
15. Children skipped meals (experience)			
16. Children skipped meals (frequency)			
17. Children were hungry but did not eat because could not afford			
18. Children did not eat for whole day			

FS: Food security group, FI: Food insecurity group, defined as a score of ≥3 according to the 18-item assessment questionnaire. ^1^ FI scores without hunger. ^2^ FI scores with hunger.

**Table 2 ijerph-20-06695-t002:** General characteristics of study subjects by food insecurity status.

	FS (*n* = 2891)		FI (*n* = 362)		*p*-Values
Gender					0.366
Boys	1525	(52.3)	199	(55.2)	
Girls	1366	(47.7)	163	(44.8)	
Residential area					0.785
Urban	2428	(84.7)	302	(85.5)	
Rural	463	(15.3)	60	(14.5)	
Household income level ^1^					<0.001
Low	207	(7.8)	121	(33.5)	
Mid-low	687	(23.9)	150	(44.3)	
Mid-high	994	(35.1)	71	(18.3)	
High	988	(33.1)	16	(3.9)	
National Basic livelihood security					<0.001
Past/current-recipient	124	(4.8)	114	(33.9)	
Non-recipient	2766	(95.2)	248	(66.1)	
Family size					0.003
1–2	95	(3.2)	27	(7.7)	
3–5	2626	(90.3)	301	(82.5)	
≥6	170	(6.5)	34	(9.8)	
Householder’s sex					<0.001
Male	1931	(72.0)	167	(47.0)	
Female	715	(28.0)	172	(53.0)	
Householder’s ages					<0.001
<50	2095	(77.2)	218	(62.0)	
50–64	430	(18.1)	75	(23.1)	
≥65	121	(4.7)	46	(14.9)	
Householder’s Marital status					<0.001
Single or Married single ^2^	221	(9.0)	119	(37.9)	
Married	2421	(91.0)	219	(62.1)	
Householder’s education level					<0.001
≤Elementary school	119	(5.5)	67	(26.3)	
Middle school	123	(6.3)	44	(12.8)	
High school	885	(39.9)	149	(48.2)	
≥College	1173	(48.3)	40	(12.8)	
Householder’s job status					<0.001
Unemployed	314	(13.3)	88	(34.0)	
Employed	1978	(86.7)	214	(66.0)	
Householder’s Health insurance type					<0.001
Regional health insurance	764	(31.1)	124	(38.9)	
Employer-provided health insurance	1833	(67.2)	141	(40.6)	
Medical aid	43	(1.7)	68	(20.5)	
Householder’s Unmet health care needs ^3^					0.003
Yes	205	(10.2)	55	(18.7)	
No	2098	(89.8)	248	(81.3)	
Householder’s Reason of Unmet health care needs					<0.001
No available time	109	(52.9)	7	(13.8)	
Light symptoms	38	(15.4)	11	(23.3)	
Financial reason	34	(19.0)	34	(58.8)	
Others	24	(12.7)	3	(4.0)	

*n* (%), *p* values are determined by χ2 test. ^1^ Household income level: low, 1st quartile; mid-low, 2nd quartile; mid-high, 3rd quartile; high, 4th quartile., ^2^ Married Single: Married (Divorced, separated or widowed), ^3^ Unmet health care needs in recent 1 year. FS: Food security group; FI: Food insecurity group, defined as a score of ≥3 according to the 18-item household food security assessment questionnaire.

**Table 3 ijerph-20-06695-t003:** Health-related characteristics of Korean adolescents by food insecurity status.

Variables	Boys			Girls		
	FS (*n* = 1524)	FI (*n* = 199)	*p*-Values	FS (*n* = 1365)	FI (*n* = 161)	*p*-Values
Height (cm) ^1^	164.7 ± 0.2	163.7 ± 0.7	0.145	161.4 ± 0.3	160.4 ± 1.3	0.453
Weight (kg) ^1^	66.9 ± 0.6	64.4 ± 1.4	0.110	55.9 ± 0.5	57.3 ± 1.6	0.416
BMI (Kg/m^2^) ^1^	22.4 ± 0.2	21.8 ± 0.4	0.215	21.4 ± 0.2	22.2 ± 0.5	0.114
Waist circumstance (cm) ^1^	76.1 ± 0.5	73.7 ± 1.0	0.033	69.6 ± 0.4	71.8 ± 1.1	0.061
Obesity index (%) ^2^			0.056			0.097
Underweight	122 (8.2)	24 (10.4)		98 (6.8)	9 (6.0)	
Normal	1219 (78.5)	164 (83.0)		1148 (84.1)	126 (78.8)	
Overweight/Obesity	183 (13.3)	11 (6.6)		119 (9.1)	26 (15.2)	
Blood parameters ^1^						
HbA1c (%)	5.4 ± 0.0	5.4 ± 0.0	0.569	5.4 ± 0.0	5.4 ± 0.1	0.934
FBS (mg/dL)	90.6 ± 0.4	89.0 ± 0.9	0.076	88.5 ± 0.4	90.4 ± 1.3	0.153
TC (mg/dL)	154.3 ± 1.4	145.6 ± 2.5	0.003	165.1 ± 1.5	161.0 ± 3.3	0.266
HDL (mg/dL)	48.0 ± 0.4	48.2 ± 0.9	0.824	55.0 ± 0.6	50.9 ± 1.2	0.003
LDL (mg/dL)	96.0 ± 2.5	87.5 ± 5.0	0.132	95.8 ± 2.7	101.0 ± 9.2	0.588
TG (mg/dL)	92.5 ± 2.9	83.6 ± 4.2	0.086	78.8 ± 2.0	84.6 ± 8.1	0.479
Subjective health status			0.005			<0.001
Good	1037 (66.3)	117 (52.7)		891 (63.2)	70 (40.5)	
Normal	406 (29.2)	69 (41.1)		418 (33.3)	75 (51.0)	
Bad	59 (4.5)	13 (6.2)		46 (3.6)	15 (8.5)	
Subjective body image			0.191			0.012
Thin	462 (31.1)	74 (35.4)		304 (20.5)	23 (13.7)	
Normal	551 (35.9)	77 (38.7)		663 (48.6)	70 (43.3)	
Fat	484 (33.0)	48 (25.9)		383 (30.8)	67 (43.0)	
Perceived Stress			0.835			0.028
More	208 (20.9)	34 (21.7)		258 (27.4)	55 (39.0)	
Less	870 (79.1)	124 (78.3)		731 (72.6)	75 (61.0)	
Melancholy ^3^			0.927			0.002
Yes	71 (7.4)	10 (7.6)		84 (9.0)	24 (18.5)	
No	1007 (92.6)	148 (92.4)		905 (91.0)	106 (81.5)	
Suicidal attempt			0.055			0.043
Yes	2 (0.5)	3 (2.5)		14 (1.5)	7 (4.4)	
No	848 (99.5)	127 (97.5)		804 (98.5)	102 (95.6)	
Psychological counseling ^4^			0.002			0.538
Yes	14 (2.5)	8 (10.2)		37 (6.8)	8 (9.0)	
No	671 (97.5)	75 (89.8)		591 (93.2)	74 (91.0)	
Smoking status			0.004			0.004
Current smoker	79 (10.7)	27 (24.3)		18 (2.4)	8 (9.4)	
Former smoker	25 (3.1)	1 (1.6)		9 (1.8)	1 (1.0)	
Non smoker	903 (86.3)	116 (74.0)		926 (95.7)	113 (89.7)	
Alcohol consumption			0.007			0.871
Yes	234 (29.9)	57 (43.4)		188 (25.1)	28 (25.9)	
No	737 (70.1)	93 (56.6)		752 (74.9)	88 (74.1)	

Data was described as mean ± SE or *n* (%) and *p*-values were determined by Student’s *t*-test or χ2 test. ^1^ (health related continuous variable) adjusted by age, ^2^ Percentile of sex-specific BMI for age growth charts: underweight (BMI < 5th percentile), normal weight (5th percentile ≤ BMI < 85th percentile), overweight and obesity (85th percentile ≤ BMI), ^3^ Continuous melancholy for over 2 weeks, ^4^ Experience in psychological counseling for the last 1 year. FS: Food security group; FI: Food insecurity group, defined as a score of ≥3 according to the 18-item household food security assessment questionnaire.

**Table 4 ijerph-20-06695-t004:** Dietary Lifestyles of Korean adolescents by food insecurity status.

Variables	Boys			Girls		
	FS (*n* = 1650)	FI (*n* = 221)	*p*-Values	FS (*n* = 1500)	FI (*n* = 184)	*p*-Values
Skipping meals						
Breakfast	401 (26.9)	79 (37.4)	0.010	387 (29.2)	66 (37.8)	0.028
Lunch	120 (8.6)	23 (11.4)	0.291	113 (9.0)	23 (14.3)	0.039
Dinner	77 (4.8)	12 (6.0)	0.513	100 (7.2)	16 (10.3)	0.226
Frequency of eating breakfast			0.001			<0.001
≥5/week	871 (65.7)	95 (48.3)		787 (65.6)	72 (51.9)	
3–4/week	150 (12.0)	31 (18.2)		161 (14.5)	17 (11.4)	
≤2/week	261 (22.3)	49 (33.5)		213 (19.9)	49 (36.7)	
Breakfast with companion			0.994			0.725
Yes	733 (68.4)	89 (68.4)		661 (67.3)	59 (65.0)	
Frequency of eating out			0.729			0.113
≥5/week	1599 (96.2)	215 (95.4)		1439 (95.2)	179 (96.2)	
1–4/week	40 (2.8)	3 (2.7)		48 (4.0)	3 (1.5)	
<1/week	11 (1.0)	3 (1.8)		12 (0.8)	2 (2.3)	
Nutrition Fact Usage			0.349			0.877
Yes	329 (22.5)	35 (18.8)		400 (32.6)	64 (33.3)	
No	1113 (77.5)	146 (81.2)		950 (67.4)	103 (66.7)	
Nutrition label effect			0.919			0.792
Yes	179 (55.8)	19 (56.8)		275 (71.6)	43 (73.3)	
No	149 (44.2)	16 (43.2)		123 (28.4)	21 (26.7)	

Data was described as *n* (%) and *p* values were determined by χ2 test. FS, Food security group; FI, Food insecurity group, defined as a score of ≥3 according to the 18-item household food security assessment questionnaire.

**Table 5 ijerph-20-06695-t005:** Nutrition intakes and dietary quality according to food insecurity status.

Variables	Boys			Girls		
	FS (*n* = 1650)	FI (*n* = 221)	*p*-Values	FS (*n* = 1500)	FI (*n* = 184)	*p*-Values
Energy intake (kcal/day) ^1^	2319.9 ± 24.1	2257.6 ± 71.2	0.410	1876.9 ± 21.9	1863.9 ± 69.0	0.858
% of total energy ^2^						
Carbohydrate	60.9 ± 0.3	61.9 ± 0.7	0.231	61.3 ± 0.3	64.1 ± 0.8	0.002
Protein	14.9 ± 0.1	14.2 ± 0.3	0.039	14.5 ± 0.1	13.5 ± 0.4	0.016
Fat	24.2 ± 0.3	23.9 ± 0.6	0.629	24.2 ± 0.3	22.4 ± 0.7	0.015
NAR ^3^						
Protein	0.96 ± 0.00	0.94 ± 0.01	0.194	0.94 ± 0.00	0.93 ± 0.01	0.186
Vitamin A ^2^	0.50 ± 0.01	0.46 ± 0.02	0.078	0.51 ± 0.01	0.51 ± 0.03	0.966
Thiamine	0.95 ± 0.00	0.94 ± 0.01	0.571	0.92 ± 0.01	0.92 ± 0.02	0.769
Riboflavin	0.83 ± 0.01	0.80 ± 0.02	0.184	0.86 ± 0.01	0.82 ± 0.02	0.062
Niacin	0.85 ± 0.01	0.83 ± 0.02	0.280	0.80 ± 0.01	0.75 ± 0.02	0.031
Vitamin C	0.56 ± 0.01	0.53 ± 0.02	0.202	0.53 ± 0.01	0.52 ± 0.03	0.835
Calcium	0.55 ± 0.01	0.53 ± 0.02	0.379	0.52 ± 0.01	0.49 ± 0.02	0.188
Phosphorus	0.89 ± 0.01	0.89 ± 0.01	0.708	0.86 ± 0.01	0.85 ± 0.01	0.386
Iron	0.83 ± 0.01	0.83 ± 0.02	0.869	0.74 ± 0.01	0.69 ± 0.02	0.051
MAR ^4^	0.77 ± 0.01	0.75 ± 0.01	0.182	0.74 ± 0.01	0.72 ± 0.01	0.160
INQ ^5^						
Protein	1.81 ± 0.01	1.75 ± 0.03	0.151	1.56 ± 0.01	1.47 ± 0.04	0.039
Vitamin A ^6^	0.68 ± 0.04	0.53 ± 0.03	0.002	0.64 ± 0.02	0.60 ± 0.04	0.410
Thiamine	1.71 ± 0.02	1.76 ± 0.05	0.254	1.53 ± 0.02	1.56 ± 0.05	0.552
Riboflavin	1.16 ± 0.01	1.11 ± 0.03	0.103	1.23 ± 0.02	1.19 ± 0.04	0.353
Niacin	1.22 ± 0.01	1.19 ± 0.03	0.391	1.05 ± 0.01	0.97 ± 0.04	0.032
Vitamin C	0.85 ± 0.03	0.74 ± 0.05	0.052	0.81 ± 0.03	0.94 ± 0.12	0.280
Calcium	0.62 ± 0.01	0.64 ± 0.03	0.645	0.57 ± 0.01	0.53 ± 0.02	0.087
Phosphorus	1.25 ± 0.01	1.30 ± 0.02	0.056	1.14 ± 0.01	1.13 ± 0.03	0.668
Iron	1.18 ± 0.04	1.28 ± 0.10	0.378	0.89 ± 0.01	0.85 ± 0.05	0.487

Data was described as mean ± SE or *n* (%) and *p* values were determined by Student’s *t*-test or χ2 test. ^1^ Energy intake (kcal/day) adjusted by age. ^2^ % of total energy adjusted by age and energy intake. AMDRs criteria: carbohydrate = 55–65% energy intake; protein = 7–20% energy intake; fat = 15–30% energy intake. ^3^ The NAR is the ratio of an individual’s intake to the age- and sex-specific Recommended nutrient intake (RI) using Dietary Reference Intakes for Koreans 2010 and 2015. ^4^ The unit used for vitamin A in the Dietary Reference intakes for Koreans (KDIRs) was changed from μg retinol equivalents (μg RE) to μg retinol activity equivalents (μg RAE) in 2015. Therefore, we used the vitamin A (μg RAE). ^5^ The MAR is calculated by averaging all the NAR values. ^6^ The INQ is the ratio of an individual’s intake per 1000 kcal to the age- and sex-specific Recommended nutrient intake (RI) per 1000 kcal using Dietary Reference Intakes for Koreans 2010 and 2015. Food security group; FI, Food insecurity group, defined as a score of ≥3 according to the 18-item household food security assessment questionnaire.

## Data Availability

The datasets analyzed in this study are available in publicly repository: http://knhanes.cdc.go.kr (accessed on 20 February 2021) (KNHANES). We provided analyzed secondary data with five tables and two figures with one Appendix A in this manuscript.

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
