# Peer review of "The Health and Nutritional Status of Children (10–18 years) Belonging to Food Insecure Households: The Korea National Health and Nutrition Examination Survey (2012–2019)"

_ijerph, 2023, doi:10.3390/ijerph20176695_

Round 1

Reviewer 1 Report

General comments:

The authors reviewed the Health and Nutrition Status of Children (10-18 years) belonging to Food Insecurity Households from 2012-2019. The manuscript can be accepted after minor revisions. Moderate English editing and spell check is required.  A few typographical errors also need to be addressed.

Comments:

Abstract: There should be no objectives, methods, or conclusions in the abstract.

Introduction: In the introduction section, the authors should highlight the importance of the present study with supporting literature. The introduction seems more like a review.

L77: [Figure 1].

L94: (with hunger, 13–18) [19].

L96: [18] [Table 1].

L199: individuals [Table 2].

L209: drinking rate [Table 3]. Rectify the spacing errors throughout the manuscript.

Discussion: Can be improved with the latest references supporting the findings of the present study.

Conclusions: It is missing.

Moderate English editing and spell check is required.  A few typographical errors also need to be addressed.

Author Response

I attached answer sheet from reviewer 2's Qustions. 

Reviewer 2 Report

I have gone through the manuscript: Health and Nutrition Status of Children (10-18 years) belong to Food Insecurity Household: Preregistered data from KNHANES 2012–2019. Overall, it is well written, I have a few minor suggestions to improve it further.

1.      The introduction does not adequately explain the association between food insecurity and children's health and nutrition status (10-18 years). It would be preferable if some relevant statistics data were also included.

2.      Several spots in the manuscript have spacing issues, such as line 77, line 145, line 209, and so on.

3.      In Discussion section, the authors may add a statement on role of family influence on habits and behavior of children, citing 10.3390/ijerph17082868.

4.      The conclusion does not appear as a distinct heading in the article. Following the discussion, the key results and their limitations/implications of the work should be addressed individually under a distinct title.

Minor improvements required

Author Response

Answer for reviewer 2

I have gone through the manuscript: Health and Nutrition Status of Children (10-18 years) belong to Food Insecurity Household: Preregistered data from KNHANES 2012–2019. Overall, it is well written, I have a few minor suggestions to improve it further.

  1. The introduction does not adequately explain the association between food insecurity and children's health and nutrition status (10-18 years). It would be preferable if some relevant statistics data were also included.

Ans>  In the introduction, we reviewed the issues of FS & FI as followings, however, we reinforced the “Introduction” with additional studies.

  • Definition of FS and FI.
  • Which population groups are vulnerable to FI? We focused on children and adolescents who are vulnerable to FI to provide fundamental data to help promote the health of vulnerable populations.
  • What kind of health environments and nutrition are associated with children belonging FI households?
  1. Several spots in the manuscript have spacing issues, such as line 77, line 145, line 209, and so on.

ANs> I rectify the spacing errors throughout the manuscript.

  1. In Discussion section, the authors may add a statement on role of family influence on habits and behavior of children, citing 10.3390/ijerph17082868.

Ans> Thank you for your advice, however, I can’t find the association between tobacco users of family and household Food Insecurity. Moreover, it was hard to explain their influence because smoking was out of questionnaires to decide FI in this study.    

  1. The conclusion does not appear as a distinct heading in the article. Following the discussion, the key results and their limitations/implications of the work should be addressed individually under a distinct title.

Ans> We totally rearranged the statement of “Discussion part” as your suggestion. We inserted “Conclusion section” which is not recognized as mandatory.

Reviewer 3 Report

Well done! 

This is a very well written manuscript. 

My suggestions are very minor. 

1. The purpose statement state 'an aim', this indicates that there was more than one aim to the research. I did not recognize the additional aim(s). If there were not additional aims, please reward to state 'the aim...'.

2. There are several uses of the word 'we' in the manuscript. My suggestion is to broaden the scope of the language to exclude the researchers specifically.  For example: Line 254 states...we included children and adolescents... A revised version may state, children and adolescents were included....

Otherwise, I found the document to be very strong and extremely close to recommend for publication. 

Please see my comments meant for the authors. 

Author Response

  • I appreciated your comments and agreed with your suggestion.

Reviewer 4 Report

Please find my comments below. I think this will help the authors improve the quality of this manuscript.

The title should be reconsidered and avoid the use of abbreviations like KNHANES.

The abstract should also have a couple introductory sentences and be written in an unstructured way (Check journal guidelines). Moreover, authors have used a number of acronyms. First, try to avoid their use in the abstract. Second, if it’s absolutely necessary, define all of them in the first place. Add some concluding sentences in the abstract as well.

Authors have limited their introduction section to only their country. Please add some international studies or perspectives to make this study interesting for international readers as well. The following studies might assist authors in achieving this goal:

https://doi.org/10.1108/BFJ-05-2021-0464

https://doi.org/10.1007/s12571-022-01312-w

Please mention the research gap that this study intends to fill in the literature. Also mention the hypotheses and possible beneficiaries of this study.

I would suggest adding a section on the review of literature after the introduction section to incorporate relevant literature.

When I look at the sample size of both the FI and FS groups, I find that the number of respondents in the FI group is only 362 compared to 2891 in the FS group. Authors should use some analytical technique to compare similar people because there should be something common when you compare two groups. Otherwise, the results would be meaningless. We all know that financially secure people have low incomes, low education, etc.

Also identify the factors responsible for FI and FS by using some econometric models.

Table 4 and 5 should be placed before the Discussion section.

Authors should discuss the results without repeating their findings in the discussion section.

More literary support is needed to justify this study.

Author Response

Please find my comments below. I think this will help the authors improve the quality of this manuscript.

1) The title should be reconsidered and avoid the use of abbreviations like KNHANES.

--> Corrected

2) The abstract should also have a couple introductory sentences and be written in an unstructured way (Check journal guidelines). Moreover, authors have used a number of acronyms. First, try to avoid their use in the abstract. Second, if it’s absolutely necessary, define all of them in the first place. Add some concluding sentences in the abstract as well.

--> Corrected

3) Authors have limited their introduction section to only their country. Please add some international studies or perspectives to make this study interesting for international readers as well. The following studies might assist authors in achieving this goal:

  • Thank you for your advice, however, we exemplified the international papers mentioned FI and children health. Since DADA in Ref# 18 with META analysis was collected by several countries including developing/developed countries such as Malaysia, Iran, Ethiopoa, Indonesia, Iran, USA and Canada, I explained the results in detail by adding other new references.
  • We rearranged the statement to be understandable, and https://doi.org/10.1007/ s12571-022-01312-w you recommended was referenced in the statement for COVID restrictive effects.

4) Please mention the research gap that this study intends to fill in the literature. Also mention the hypotheses and possible beneficiaries of this study.

  • I mentioned our purposes of the study were to analyze the Nutritional Health status of children and adolescents who are vulnerable to FI, therefore, we want to analyze the environmental data for children’s nutrition-health with FI to improve educational and policymaking programs in the future. (L67-72)

5) I would suggest adding a section on the review of literature after the introduction section to incorporate relevant literature.

  • This study is exploring the importance of children’ health with FI using the Secondary DATA of KNHANES, not manuscript for literature review. I referenced a well-reviewed paper, “ Am J Clin Nutr 2014;100:684–92. & Paediatr Child Health.2015 Mar; 20(2): 89–91” and supplemented other 10 studies in “Introduction & Discussion” part.  
  •  

6) When I look at the sample size of both the FI and FS groups, I find that the number of respondents in the FI group is only 362 compared to 2891 in the FS group. Authors should use some analytical technique to compare similar people because there should be something common when you compare two groups. Otherwise, the results would be meaningless. We all know that financially secure people have low incomes, low education, etc.

-->  All data in Table 2 came from X2 test of independence that shows how the frequencies for each factor differ in two categories of FS and FI. Therefore, statistical significance of frequencies in the levels of income, material status, education, job and healthcare status according to FS and FI were mentioned. Similar frequencies between FS and FI were not mentioned.

-->  Yes, I agreed that financially secure people have low incomes, low education, etc. but I mentioned Food Insecurity households judged by FS/FI assessment were associated with not only unstable socioeconomic environments, such as low income, unemployment, single household, but low education. Additionally, I do not want to say that I was caused by the above factors as results in X2 test.  

7) Also identify the factors responsible for FI and FS by using some econometric models.

--> It was hard to show the econometric models because 1st , we used preregistered secondary data that some of them could not be manipulated for the model, and 2nd , many influencing diverse factors including both continuous and categorical data were differently involved in FS or FI, respectively. Without it, we can explain what factors including nutritional health may affect FS and FI in children. In the future, we will consider how to use it. 

8) Table 4 and 5 should be placed before the Discussion section.

  • Corrected

9) Authors should discuss the results without repeating their findings in the discussion section.

  • We totally rearranged the statement of “Discussion part” as your suggestion.

More literary support is needed to justify this study.

Round 2

Reviewer 4 Report

Satisfied with authors responses and amendments in revised manuscript.